# An *Escherichia coli*-Based Phosphorylation System for Efficient Screening of Kinase Substrates

**DOI:** 10.3390/ijms25073813

**Published:** 2024-03-29

**Authors:** Andrés Cayuela, Adela Villasante-Fernández, Antonio Corbalán-Acedo, Elena Baena-González, Alejandro Ferrando, Borja Belda-Palazón

**Affiliations:** 1Instituto de Biología Molecular y Celular de Plantas (IBMCP), Consejo Superior de Investigaciones Científicas, Universitat Politècnica de València, 46022 Valencia, Spain; cayuelaandres@gmail.com (A.C.); avilfer2@ibmcp.upv.es (A.V.-F.); acorace@ibmcp.upv.es (A.C.-A.); 2Department of Biology, University of Oxford, Oxford OX1 3RB, UK; elena.baena-gonzalez@biology.ox.ac.uk

**Keywords:** posttranslational modifications, protein phosphorylation, kinase, *Escherichia coli* phosphorylation system, *Arabidopsis thaliana*, SnRK1 signaling network, mitogen-activated protein kinase cascades

## Abstract

Posttranslational modifications (PTMs), particularly phosphorylation, play a pivotal role in expanding the complexity of the proteome and regulating diverse cellular processes. In this study, we present an efficient *Escherichia coli* phosphorylation system designed to streamline the evaluation of potential substrates for *Arabidopsis thaliana* plant kinases, although the technology is amenable to any. The methodology involves the use of IPTG-inducible vectors for co-expressing kinases and substrates, eliminating the need for radioactive isotopes and prior protein purification. We validated the system’s efficacy by assessing the phosphorylation of well-established substrates of the plant kinase SnRK1, including the rat ACETYL-COA CARBOXYLASE 1 (ACC1) and FYVE1/FREE1 proteins. The results demonstrated the specificity and reliability of the system in studying kinase-substrate interactions. Furthermore, we applied the system to investigate the phosphorylation cascade involving the *A. thaliana* MKK3-MPK2 kinase module. The activation of MPK2 by MKK3 was demonstrated to phosphorylate the Myelin Basic Protein (MBP), confirming the system’s ability to unravel sequential enzymatic steps in phosphorylation cascades. Overall, this *E. coli* phosphorylation system offers a rapid, cost-effective, and reliable approach for screening potential kinase substrates, presenting a valuable tool to complement the current portfolio of molecular techniques for advancing our understanding of kinase functions and their roles in cellular signaling pathways.

## 1. Introduction

Posttranslational modifications (PTMs) are proteolytic or covalent modifications that occur on amino acid side chains of the proteins or at their N- or C- termini [1]. PTMs, enzymatically catalyzed in a reversible or irreversible manner, increase the complexity of the proteome by expanding the number of possible proteoforms, serving as essential mechanisms used by eukaryotic cells to diversify their protein functions and dynamically regulate diverse cellular processes [1,2,3]. Indeed, these modifications play a crucial role in regulating the structure and function of proteins, affecting their activity, cellular localization, interactions, and stability [1].

Among the wide variety of PTMs, phosphorylation is the most widespread type of PTM used in signal transduction and in the reversible regulation of most metabolic and physiological pathways in eukaryotes [4,5]. Phosphorylation is catalyzed by one of the largest families of genes in eukaryotes, the protein kinases, which transfer the γ-phosphate from ATP to specific amino acids, mainly serine, threonine, and tyrosine residues, in the target proteins. It is well-established that the recognition of the amino acid sequence context (phosphorylation motif) surrounding these residues is determinant for the binding and consequent phosphorylation reaction [6].

The broad importance of phosphorylation in eukaryotic biology is underscored by the substantial number of kinase-coding genes in eukaryotic genomes. For example, out of 20,279 protein-coding genes in the human genome (neXtprot 15 February 2021 release), 518 are protein kinase genes (2.5%) [7]. Notably, the model plant *Arabidopsis thaliana*, with 27,416 protein-coding genes (TAIR10) [8], significantly surpasses this number, with 942 genes encoding kinases (3.4%) [9]. This is in accordance with the large expansion and functional diversification observed in many other plant gene families and is thought to have provided an advantage for a sessile lifestyle [10]. Despite the existent catalogue of technologies to characterize kinase substrates, there is still room for simple, quick, and affordable approaches to achieve this goal [11,12,13,14]. Kinase targets are typically evaluated through in vitro kinase assays that involve the purification of recombinant proteins or immunoprecipitation (IP) of the kinase under study and the use of radioactivity. This kind of experimental approach can face limitations and is not free of drawbacks, as purification of recombinant proteins can be difficult in some cases. IP, on the other hand, relies on the availability of protein-specific antibodies that are not always easy to obtain. The use of epitope tags can overcome this problem [15], but the epitope in turn may alter the activity or the specificity of the kinase. In this context, the expansion of the currently available methods should facilitate the study of phosphorylation with minimum manipulation of cell extracts and rapid, sensitive, and easy analysis of the phosphorylation reaction.

To illustrate how important it is to expand the available tools to study the PTMs by phosphorylation and to complete the characterization of kinase substrates in plants, we have focused on two signaling modules involved in the response to both abiotic and biotic stresses, which facilitate the integration of hormonal and environmental signaling during stress responses [16,17]. The first module is the SNF1-related kinase (SnRK) superfamily, encompassing SnRK1, SnRK2, and SnRK3 subgroups, which serves as a central regulator of plant growth, development, and tolerance to diverse abiotic stresses [18,19,20,21]. Over the past years, research in the field has been particularly focused on the study of SnRK1, the plant ortholog of the animal AMP-activated protein kinase (AMPK) and SNF1 in yeast, given its role as a major regulatory hub. SnRK1 senses and integrates the energy status with very diverse stress conditions, including those involving abscisic acid (ABA), and is crucial in maintaining energy homeostasis for growth and survival [21,22,23,24,25]. SnRK1 is a heterotrimeric complex comprising an α-catalytic subunit and regulatory β and βγ subunits encoded by various genes in *A. thaliana*: *SnRK1α1/α2/α3*, *SnRK1β1/β2/β3*, and *SnRK1βγ* [26]. Despite the generally accepted role of SnRK1 as a central mediator in energy and stress signaling pathways, a comprehensive understanding of the SnRK1 signaling network is still missing. Thus, the identification of direct targets (enzymes and transcription factors, among others) that link SnRK1 to various cellular and physiological processes in the plant is crucial for understanding how this central kinase impacts development and growth in relation to stress responses [27]. In this context, the protein interaction landscape, composition, and structure of the SnRK1 heterotrimer have recently provided valuable insights through the combination of affinity purification, proximity labeling, and crosslinking mass spectrometry [28].

The second kinase group relevant for plant development and stress responses is composed of the members of the Mitogen-activated protein kinase (MAPK) cascades [29,30,31]. A typical MAPK cascade consists of at least one MAPK (MPK), one MAPK kinase (MAPKK, MKK, or MEK), and one MAPKK kinase (MAP3K, MKKK, or MEKK). Upon stimulation, generally by extracellular signals, the activation of top-tier MAP3K(s) leads to the phosphorylation and activation of downstream MKK(s). These activated MKK(s) subsequently phosphorylate and activate the bottom-tier MPK(s). The activated MPKs then phosphorylate various downstream substrates, such as transcription factors, protein kinases, enzymes, and structural proteins, ultimately triggering cellular responses [31,32]. Furthermore, these kinases can assemble in various combinations that play different roles in biotic and abiotic stresses, as well as in cell division and development [17].

Among these combinations or regulatory modules, the one composed of MAP3K17/18, MKK3, and the MPK C-type MPK1/2/7/14 has been implicated in ABA signaling. The activation of this module could play a role in generating a robust signal to induce ABA-dependent responses under persistent stress conditions [33,34]. While the specific targets of these MPKs are yet to be discovered, phosphoproteomics studies have revealed numerous ABA-induced phosphosites in transcription factors and protein kinases, indicating their potential involvement as MAPK targets [35,36]. Hence, rigorous confirmation of the putative substrates for this regulatory module is imperative.

In the present work, we report a novel system designed to facilitate the testing of phosphorylation of potential SnRK1 substrates as well as putative targets of the MKK3-MPK2 module of *A. thaliana*. Employing this methodology for assessing phosphorylations eliminates the need to purify recombinant proteins and to use radioactive isotopes, thus increasing the current catalogue of methods and technologies for phosphorylation studies [14].

## 2. Results

### 2.1. Development of a Methodology for Analyzing the Phosphorylation of Potential SnRK1 Substrates in Escherichia coli Cells Co-Expressing Both the Kinase and the Substrate in the Same Vector

The technical limitations to perform in vitro phosphorylation assays prompted us to develop a rapid and reliable method for the evaluation of potential phosphorylations without the requirement to purify both the enzymatic components (kinases) and the substrates (targets) of the phosphorylation reactions. The conceptual strategy was based on the use of *E. coli* as a heterologous system to co-express the kinase/s and target by cloning the corresponding genes in the same IPTG-inducible expression vector and fused to distinct tags (His and/or GST). During the IPTG induction, the *E. coli* cells are expected to produce an equimolar amount of the kinase and the potential substrate. Subsequently, phosphorylation could be rapidly evaluated from a total protein extract by Western Blot (WB), either by using available antibodies against the specific phospho-protein, or by means of anti-His or anti-GST antibodies after Phos-tag SDS-PAGE separation [37], which delays the migration of the phosphorylated proteins. We have named this method as *Phoscreen*, and its schematic protocol is shown in Figure 1.

To validate this methodology, we first focused on the master regulatory kinase SnRK1 of *A. thaliana*. We used two variants of the α1 catalytic subunit of SnRK1 (SnRK1α1), either harboring a mutation that mimics phosphorylation of the activation loop (T-loop, T175D), rendering the kinase constitutively active, or harboring a mutation in the ATP-binding pocket that renders the kinase inactive [38]. As a SnRK1 direct target (positive control), we took advantage of a rat ACETYL-COA CARBOXYLASE 1 (ACC1) peptide (S55-V108), harboring the S79 residue that has been reported to be phosphorylated by SnRK1 [25,39,40]. As a negative control, we also used a non-phosphorylatable mutated version of this peptide, ACC1^S79A^. The different genetic variants of SnRK1α1 and ACC1 were cloned into *E. coli* IPTG-inducible pQLink vectors fused to His and GST, respectively [41]. Afterwards, the constructs were subcloned together in the same vector by Ligase Independent Cloning (LIC) in the following combinations: His-SnRK1α1^T175D^ + GST-ACC1; His-SnRK1α1^T175D^ + GST-ACC1^S79A^, His-SnRK1α1^K48M^ + GST-ACC1.

As shown in Figure 2a, correct protein expression of the constructs was confirmed through the separation of the protein extracts by SDS-PAGE and Coomassie staining, visualizing the fusion proteins of the expected molecular weights. As expected, only the construct containing the constitutively active version of SnRK1α1 (SnRK1α1^T175D^) was able to phosphorylate ACC1 at S79, shown by immunoblot with anti-phospho-S79-ACC1 antibodies (Figure 2b). In contrast, *E. coli* cultures expressing the kinase dead variant of SnRK1α1 (SnRK1α1^K48M^) or the non-phosphorylatable ACC1 (ACC1^S79A^) did not phosphorylate ACC1 at S79. Despite the presence of the phosphorylated S79 of ACC1 in *E. coli* cells expressing SnRK1α1^T175D^, the Phos-tag SDS-PAGE did not show any delay in the migration of the anti-GST immunogenic band.

In addition to the use of a well-reported substrate of SnRK1 as the synthetic ACC peptide, we aimed to analyze the phosphorylation of an endogenous canonical *A. thaliana* SnRK1 reported substrate, the FYVE1/FREE1 (FYVE DOMAIN PROTEIN REQUIRED FOR ENDOSOMAL SORTING 1) protein shown to be phosphorylated by SnRK1 at S530 [42]. The biological consequence of the SnRK1-dependent phosphorylation of FYVE1 is its sequestration to the forming phagophore to promote its closure to mature as an autophagosome, under conditions of nutritional deprivation. Therefore, we cloned the entire coding sequence of FYVE1 as a translational fusion to the GST-tag. Subsequently, we subcloned this construct for co-expression with either constitutively active (T175D) or inactive (K48M) versions of SnRK1α1 fused to His-tag in the following order: His-SnRK1α1^T175D^ + GST-FYVE1, His-SnRK1α1^K48M^ + GST-FYVE1. After verifying the co-expression of both fusion proteins after SDS-PAGE separation and Coomassie staining (Figure 2c), we could detect, by Phos-tag SDS-PAGE and anti-GST WB, the phosphorylation of the GST-FYVE1 protein only for *E. coli* cells expressing constitutively active SnRK1α1^T175D^. This was evident from the appearance of an immunogenic band with delayed migration on the Phos-tag gel, indicative of phosphorylation (Figure 2d). These results confirm the validity of our methodology for screening multiple potential SnRK1 substrates, either by means of phospho-specific antibodies or with the use of Phos-tag SDS-PAGE and GST-antibodies.

### 2.2. The Phoscreen System Proves to Be a Valid Tool for Analyzing Phosphorylation Cascades

The results obtained demonstrate that the *Phoscreen* system serves as a reliable and robust tool for analyzing substrate phosphorylation by a specific kinase. Nevertheless, we aimed to challenge the system’s performance with the test of phosphorylation cascades. To address this objective, we selected a case example with the analysis of the regulatory module comprising the *A. thaliana* MKK3 and the C-type MPK2 kinases [34]. We used a constitutively active mutated form of MKK3, harboring the S235E and T241E mutations (MKK3^EE^) [33], along with wild-type versions of MPK2. We speculated that constitutively active MKK3 could phosphorylate and subsequently activate MPK2 when co-expressed in *E. coli* upon IPTG induction. As a final readout, we used a well-known MPK2 substrate, a peptide from the bovine Myelin Basic Protein (MBP) encompassing residues V93 to Q102, which contains the T97 residue known to be phosphorylated by MPKs [33]. Following this approach, MKK3 and MPK2 were cloned into *E. coli* IPTG-inducible pQLink vectors as translational fusions to His, whereas the MBP peptide was cloned as a fusion to GST [41]. Afterwards, they were subcloned together in the same vector by LIC in the following combinations: His-MKK3^EE^ + GST-MBP; His-MPK2 + GST-MBP; His-MKK3^EE^ + His-MPK2 + GST-MBP.

As shown in Figure 3a, co-expression of all fusion proteins from the same vector in *E. coli* was confirmed through the separation of the protein extracts in SDS-PAGE and Coomassie staining. As expected, and according to the delay in the mobility of the GST-MBP after Phos-tag SDS-PAGE, only the cultures expressing constitutively active MKK3 together with MPK2 were able to phosphorylate the MBP peptide, leading to a clear delay in protein mobility. In contrast, when co-expressed alone, MKK3^EE^ or MPK1/2 were not able to phosphorylate MBP (Figure 3b). These results demonstrated that in our system, the constitutively active version MKK3^EE^ is capable of activating MPK2, which subsequently phosphorylates MBP. However, this could not rule out the possibility that the activity of MPK2, once activated, might not be substrate-specific and could perform promiscuous phosphorylation. To demonstrate the specificity of the phosphorylation by the activated MAPK cascade, we analyzed the ability of the MKK3^EE^-MPK2 regulatory module to phosphorylate the ACC1 peptide in comparison to the active version of SnRK1α1 (SnRK1α1^T175D^). To achieve this, His-MKK3^EE^, His-MPK2, and GST-ACC1 were cloned together in the same vector and their co-expression validated by separation of the protein extracts through SDS-PAGE and Coomassie staining (Figure 3c). As depicted in Figure 3d, cells expressing constitutively active MKK3 along with MPK2 did not induce phosphorylation at S79 of ACC1, in contrast to the robust phosphorylation observed at S79 due to the activity of constitutively active SnRK1α1. These results demonstrated the specificity of the studied phosphorylation events.

## 3. Discussion

Classical phosphoproteomics approaches using kinase loss-of-function/over-expression mutants have been useful for the large-scale identification of potential substrates of a given kinase [43]. Also, proximity labelling strategies generate relevant broad information about close interactors although not solving the problem of identifying specific substrates [44]. The lists of potential substrates generated through these strategies need to be validated in the laboratory through primarily in vitro phosphorylation assays [14], which may be a limiting step in many cases when the recombinant proteins need to be purified.

With this work we have established a robust and efficient *E. coli* system for screening multiple potential kinase substrates and for analyzing phosphorylation cascades. The approach involves co-expressing the enzymes and their corresponding substrates in a single expression vector. In addition, the use of this methodology to evaluate phosphorylation avoids the use of radioactive isotopes, and it does not require prior purification of the intervening proteins. After the rapid and efficient extraction of total protein extracts using the Laemmli method, they are separated and analyzed through WB assays. Depending on the availability of specific phospho-antibodies against the modified residue, the reaction can be directly monitored by WB assays after the separation of total extracts by standard SDS-PAGE (Figure 1). However, if specific antibodies are not available, the extracts are separated using Phos-tag SDS-PAGE, and immunodetection is performed with antibodies against a GST or His tag to which the candidate substrate protein has been fused (Figure 1). Thus, we can ascertain whether proteins have been phosphorylated by observing a delay in the electrophoretic mobility of the phosphorylated proteins [37]. To validate the system, we have focused on SnRK1 substrates and the MKK3/MPK2 phosphorylation cascades (Figure 2 and Figure 3).

This system is transferable to other known kinases, not only in plants but also in all eukaryotes, utilizing *E. coli* as a heterologous system in a rapid and efficient manner. Moreover, this system can be applied to any phosphorylatable substrate for which we seek to identify the kinase(s) [37]. The system is, however, not exempt of limitations. On the one hand, when no anti-phospho-antibodies are available, the reliance on the use of the Phos-tag may pose risks, as it may fail to separate certain phospho-proteins [45]. This is evident in the phosphorylation of ACC1 at S79, which could not be discerned using the Mn^+2^-based Phos-tag method applied in this study (Figure 2b). To overcome this problem, an alternative method using Zn^+2^-Phos-tag and a neutral pH buffer system could be performed [46,47]. Another drawback of the *Phoscreen* system is that, as neither the kinase nor the substrate needs to be purified, as there is no need to purify either the kinase or the substrate, it precludes the fine biochemical characterization of the enzymatic reaction to determine the catalytic parameters.

Among the advantages of this system are the following: (i) it provides the opportunity to analyze the phosphorylation of potential substrates without prior purification, allowing for a clear observation of their expression from IPTG-induced protein extracts after separation in SDS-PAGE and Coomassie staining; (ii) this, in turn, offers the possibility of excising the band corresponding to the induced substrate protein under study for subsequent analysis through LC-MS/MS to identify the specific phosphorylated residue(s). It is important to notice that being the enzymatic reaction taking place inside the *E. coli* cells, appropriate controls should always be used to monitor potential unspecific phosphorylation events by endogenous *E. coli* kinases, although we cannot rule out the remote possibility that bacterial kinases could be activated by expression of the non-bacterial kinase to subsequently phosphorylate the substrate. Nevertheless, the presence of the tags (GST or His) fused to the proteins of interest can always allow the purification of proteins by affinity chromatography to perform in vitro assays in case of suspicious unspecific phosphorylation. A further refinement of the specificity of this phosphorylation system in *E. coli* could involve the use of analog-sensitive (AS) versions of the kinases under investigation. AS kinase variants harbor a mutation in a highly conserved “gatekeeper” residue that enlarges the ATP-binding pocket. This allows the kinase to accommodate bulky ATP-mimicking inhibitors or modified ATP molecules that cannot bind to non-engineered kinases [48]. An example of the latter is N^6^-modified ATPγS, which is used by the AS kinase to transfer the thiophosphate group to specific substrates that can be later detected after SDS-PAGE separation and WB analysis by using specific thioester antibodies [49]. Also, thiophosphorylated proteins can be captured by thiol-specific chemistry and identified by LC-MS/MS, including the identification of the exact modified residue(s) [50]. In this regard, based on the results performed with AMPKα2, the human SnRK1α1 ortholog [51], the conserved “gatekeeper” residue M96 inside the SnRK1α1 ATP-binding pocket that allows the acceptance of bio-orthogonal ATP analogs was identified [52].

The system developed in this work is potentially transferable to other known PTMs since the enzymes responsible for the specific PTM and its substrate can be cloned in the same vector [41]. In this regard, our laboratory has already pioneered the use of the same system to monitor the modification of the translation factor eIF5A of *A. thaliana* by hypusination, through the sequential activity of the two enzymes DHS and DOHH, which catalyze the transfer of a 4-aminobutyl group from spermidine to a conserved lysine of eIF5A [53]. In general, PTMs can regulate and/or influence the activity of proteins and enzymes in multiple ways, which may or may not be mutually exclusive: (i) altering the structural conformation, leading to their activation, deactivation, or modification of function; (ii) changing subcellular localization; (iii) interfering with protein–protein interaction capacity; and (iv) affecting protein half-life [3]. Additionally, it is crucial for bringing about significant biotechnological improvements at the protein level through PTMs, providing enhanced biochemical characteristics to the proteins themselves, such as increased activity, solubility, and/or stability, among others. We also envisage that the methodology developed in this work could potentially be applied for the straightforward production, either in the laboratory or on an industrial scale, of post-translationally modified proteins of biotechnological interest.

## 4. Materials and Methods

### 4.1. Biological Material and Growth Conditions

For the cloning of all the constructs obtained in this work, chemically competent cells from the *E. coli* TOP10 strain were used. For heterologous protein expression experiments, chemically competent cells of BL21(DE3) pLysE strain were used. For the growth of competent *E. coli* cells, LB (Lysogeny Broth) medium was used, composed of 1% tryptone, 1% NaCl, 0.5% yeast extract. In the case of growth in solid medium, 2% bacteriological agar was added.

For the transformations, 20–100 ng of DNA was added to 50–100 µL of a concentrated suspension of competent cells and transformed by heat shock. Transformed cells were plated on solid LB containing 50 µL/mL carbenicillin and allowed to grow overnight at 37 °C.

### 4.2. Cloning Procedures

The list of entry vectors used in this work is described in Table 1. The cloning procedures carried out for the entry vectors generated in this work are described as follows.

#### 4.2.1. Generation of T175D or K48M Mutations in SnRK1α1

The T175D or K48M mutations were introduced into the SnRK1α1 coding sequence (*KIN10*) by PCR-mediated site-directed mutagenesis. First, two separated PCRs were performed using the Accuprime Pfx DNA polymerase (Thermo Fisher Scientific, Waltham, MA, USA) and the pENTR 3C-KIN10 as a template. In the case of the T175D, PCR1 used the KIN10_Fw_attB1 and KIN10_T175D_Rv primers, and PCR2 the KIN10_T175D_Fw and KIN10_Rv_attB2 ones (Table 2). In the case of the K48M, PCR1 used the KIN10_Fw_attB1 and KIN10_K48M_Rv primers, and PCR2 the KIN10_K48M_Fw and KIN10_Rv_attB2 ones (Table 2). The PCR program was the following: 95 °C for 2 min; 35× (95 °C for 15 s, 55 °C for 30 s, 68 °C for 100 s); 72 °C for 10 min. PCR products were purified by gel band purification and used as templates for PCR3 using the Phusion High–Fidelity DNA Polymerase (Thermo Fisher Scientific, Waltham, MA, USA) and the KIN10_Fw_attB1 and KIN10_Rv_attB2 primers. The PCR program was the following: 98 °C for 30 s; 5× (98 °C for 10 s, 45 °C for 30 s, 72 °C for 1 min); 25× (98 °C for 10 s, 60 °C for 30 s, 72 °C for 1 min); 72 °C for 10 min. PCR3 product was purified by gel band purification and used for BP Gateway recombination with pDONR/ZEO.

#### 4.2.2. Cloning of ACC1 Peptide (S55-V108) and Its Mutated Version (S79A)

The coding sequence of the ACETYL-COA CARBOXYLASE 1 (ACC1) peptide from S55 to V108 of *Rattus norvegicus* (Rat, P11497), which includes the S79 target residue of SnRK1, was obtained by PCR. The Phusion High–Fidelity DNA Polymerase (Thermo Fisher Scientific, Waltham, MA, USA), along with the ACC1_S55-V108_attB1 and ACC1_S55-V108_attB2 primers, was used for the amplification (Table 2). The pGEX-4T1-ACC1 construct served as the template [28]. The PCR program was the following: 98 °C for 30 s; 35× (98 °C for 10 s, 58 °C for 30 s, 72 °C for 15 s); 72 °C for 5 min. In the case of the ACC1 peptide mutated at S79 (ACC1^S79A^), the same steps were followed, but in this instance, the pGEX-4T1-ACC1_S79A construct was used as the template [28]. PCR products were purified by gel band purification and used for BP Gateway recombination with pDONR/ZEO.

#### 4.2.3. Cloning of MPK2 Coding Sequence

The *MPK2* coding sequence was amplified from a preparation of *A. thaliana* cDNA using primers containing the attB1 and attB2 sites (Table 2) and the Accuprime Pfx DNA polymerase (Thermo Fisher Scientific, Waltham, MA, USA). The PCR program was the following: 95 °C for 2 min; 35× (95 °C for 15 s, 55 °C for 30 s, 68 °C for 90 s); 72 °C for 10 min. The PCR product was purified by gel band purification and used for BP Gateway recombination with pDONR/ZEO.

#### 4.2.4. Cloning of MBP Peptide (V93-Q102) Coding Sequence

The coding sequence of the Myelin Basic Protein (MBP) peptide from V93 to Q102 of *Bos taurus* (Bovine, P02687) containing the T97 target residue of MPKs was obtained by overlap extension PCR, using the Phusion High–Fidelity DNA Polymerase (Thermo Fisher Scientific, Waltham, MA, USA) and the MBP_V93-Q102_attB1 and MBP_V93-Q102_attB2 overlapping primers (Table 2). The PCR program was the following: 95 °C for 30 s; 30× (95 °C for 10 s, 55 °C for 30 s, 72 °C for 50 s); 72 °C for 10 min. The PCR product was purified by gel band purification and used for BP Gateway recombination with pDONR/ZEO.

#### 4.2.5. BP Gateway Recombination

PCR products were cloned into pDONR/ZEO vector by overnight incubation at 25 °C of 50 ng of the PCR product, 100 ng of pDONR/ZEO, 1 µL of BP Clonase II Enzyme mix (Thermo Fisher Scientific, Waltham, MA, USA). This incubation was followed by a 10 min incubation at 37 °C with the addition of 1 µL of 1% proteinase K. After this, recombinations were used to transform TOP10 competent cells. Selection of transformed cells was carried out in solid LB medium with 50 μg/mL zeocin.

#### 4.2.6. Destination Constructs

The list of destination vectors generated in this work is depicted in Table 3, along with an explanation of the cloning method employed to produce the specific construct. In general, they were obtained by the combination of LR Gateway reactions and ligation independent cloning (LIC) techniques.

Coding sequences were cloned into pQLinkHD (N-terminal fusion to a histidine tag) or pQLinkGD (N-terminal fusion to a GST tag) by LR Gateway recombination with the corresponding entry vector [41]. To do this, 50–100 ng of each linearized entry vector was mixed with 100 ng of either pQLinkHD or pQLinkGD, and 1 µL of LR Clonase II Enzyme mix (Thermo Fisher Scientific, Waltham, MA, USA) in a final volume of 5 µL. Samples were incubated overnight at 25 °C and this incubation followed by a 10 min incubation at 37 °C with 1 µL of 1% proteinase K. After this, recombinations were used to transform TOP10 competent cells. Selection of transformed cells was carried out in solid LB medium with 50 μg/mL carbenicillin.

The LIC method used in this work was based on the assembly of PacI digestion fragments with vectors linearized with SwaI [41]. For this, 1 µg of the pQLinkG-derived destination vectors was cut with PacI (Thermo Fisher Scientific, Waltham, MA, USA), and 1 µg of the pQLinkH-derived destination vectors was cut with SwaI (Thermo Fisher Scientific, Waltham, MA, USA) following the manufacturer’s instructions. Restriction products were separated on agarose electrophoresis and the fragments of interest purified by gel band purification. After this, restriction fragments were differentially treated with T4 polymerase depending on the restriction performed:50 ng of the PacI fragment was mixed with 2.5 mM dCTP, 1 U of T4 DNA polymerase (Thermo Fisher Scientific, Waltham, MA, USA), and 3 μL of 5× reaction buffer in a final volume of 15 μL.50 ng of the SwaI fragment was mixed with 2.5 mM dGTP, 1 U of T4 DNA polymerase (Thermo Fisher Scientific, Waltham, MA, USA), and 3 μL of 5× reaction buffer in a final volume of 15 μL.

Mixtures were incubated for 20 min at 22 °C followed by 20 min at 65 °C (inactivation). For the ligation of the fragments, 10 ng of each fragment treated with T4 was mixed and incubated for 2 min at 65 °C. After this, 6 mM EDTA was added and the mix incubated overnight at 25 °C. LIC reactions were used to transform TOP10 *E. coli* competent cells. Selection of transformed cells was carried out in solid LB medium with 50 μg/mL carbenicillin.

### 4.3. Protein Analysis and Manipulation Techniques

#### 4.3.1. Heterologous Expression

Once confirmed through digestion and electrophoresis, all constructs were transformed into BL21(DE3) pLysE competent cells and selected in solid LB medium containing 50 μg/mL carbenicillin. Transformant colonies were inoculated into 3 mL of liquid LB with 50 μg/mL carbenicillin and grown overnight at 37 °C with vigorous agitation. The following day, 60 µL of the saturated cultures was inoculated into 6 mL of LB containing 50 μg/mL carbenicillin and cultures incubated for 2–3 at 37 °C in agitation (200–220 rpm) until reaching an OD 600 nm of 0.2–0.3. At this moment, 1 mL of the culture was aliquoted to a new tube to serve as the -IPTG control sample. The remaining culture was induced with IPTG by adding 1 mM IPTG directly to the media and incubated for 2.5 h at 28 °C in agitation (200–220 rpm). After the induction, 1 mL was aliquoted to a new tube as the +IPTG sample. Both −IPTG and +IPTG samples were centrifuged at 12,000× *g* for 5 min and supernatants discarded. Sedimented cells were stored at −20 °C until protein extraction.

#### 4.3.2. Protein Extraction

Collected cells were resuspended in 1× Laemmli buffer (63 mM Tris-HCl pH 6.8, 2% SDS, 10% glycerol, 0.025% Bromophenol blue, 2% β-mercaptoethanol), 100 µL in the case of the −IPTG samples and 200 µL in the +IPTG ones. Samples were vortexed vigorously and boiled at 95 °C for 15 min. After boiling samples were tempered on ice for 5 min and centrifuged at 12,000× *g* for 5 min. Finally, supernatants (total protein extracts) were collected into new tubes.

#### 4.3.3. SDS-PAGE and Coomassie Staining

Samples were separated in an SDS-PAGE using Mini-Protean^®^ Tetra handcast system (Bio-Rad, Hercules, CA, USA). The stacking gel consisted of 4% acrylamide, 0.12% SDS, 124 mM Tris-HCL pH 6.8, 0.04% APS (ammonium persulfate), 0.4% TEMED. The resolving gel consisted of 8–12% acrylamide, 376 mM Tris-HCL pH 8.8, 0.1% SDS, 0.1% APS, 0.2% TEMED.

SDS-PAGE gels were run at 90–120 V in electrophoresis buffer (25 mM Tris, 192 mM glycine, 0.1% SDS) until the front reached the end and used for either WB analysis or Coomassie Blue Staining (CBS). For CBS, gels were incubated with Imperial Coomassie R-250 (Thermo Fisher Scientific, Waltham, MA, USA) for 20 min in gentle agitation. Afterwards, gels were destained by incubating overnight with a destain solution made of 10% methanol and 10% acetic acid.

#### 4.3.4. Mn^+2^-Phos-Tag SDS-PAGE

For Phos-tag SDS-PAGE analyses, proteins were separated in an 8–10% SDS-PAGE, containing 50 μM Phos-tag™ AAL16-107 and 100 μM MnCl_2_. Gels were run at a constant voltage of 30 mA/gel in electrophoresis buffer until the front reached the end and used for WB analysis. Prior to membrane transfer, the Phos-tag gels were washed with transfer buffer (tris base 5g/L and boric acid 3.1g/L) containing 1 mM EDTA for 10 min followed by a second washing of transfer buffer for another 10 min.

#### 4.3.5. Western Blot (WB)

For immunoblotting, proteins were transferred to PVDF membranes, which were previously activated with 96% ethanol for 2 min, overnight 12–20 V at room temperature using the wet blotting transfer system of Bio-Rad (Hercules, CA, USA). For the immunological detection of the proteins of interest, the following incubation steps were carried out:1 h incubation at room temperature with blocking buffer [5% *w*/*v* non-fat dry milk in 1× TBS (Tris buffered saline, 0.05 M Tris and 0.15 M sodium chloride, pH 7.6), 0.05% Tween^®^] in agitation.Overnight incubation at 4 °C on rocking platform with the primary antibody of interest (Table 4) diluted in blocking buffer.3 washes of 10 min with 1× TBS containing 0.05% Tween^®^.1 h incubation at room temperature on rocking platform with the corresponding secondary antibody (Table 4).3 washes of 10 min with 1× TBS containing 0.05% Tween^®^.

The immunological detection was performed using complementary chemiluminescent detection substrates depending on the required sensitivity: SuperSignal™ West Pico PLUS Chemiluminescent Substrate (Thermo Fisher Scientific, Waltham, MA, USA) and ECL™ Select Western Blotting Detection Reagent (Cytiva, Marlborough, MA, USA). Images were acquired using ChemiDoc system (Biorad, Hercules, CA, USA) equipped with a CCD camera.

## 5. Conclusions

We have developed a methodology to monitor phosphorylation events using *E. coli* as a heterologous system in a rapid and efficient manner by co-expressing the enzymes and the corresponding substrates in a single expression vector. The validation of the system was carried out using *A. thaliana* SnRK1α1 in *E. coli* cells expressing both the kinase and its known substrates ACC1 and FYVE1. Additionally, this approach has been successfully validated for the analysis of phosphorylation cascades, such as the one formed by the MKK3-MPK2 module and their canonical substrate MBP.

## Figures and Tables

**Figure 1 ijms-25-03813-f001:**
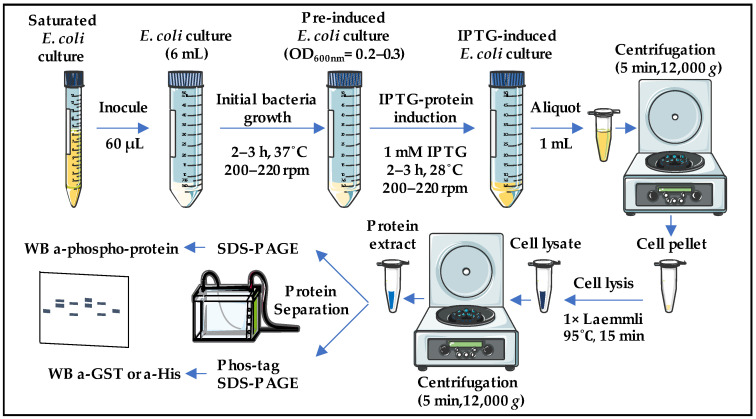
The workflow of the *Phoscreen* system. A 6 mL *E. coli* culture is treated with IPTG to induce co-expression of both GST or His-tagged kinase and substrates. After collecting the cells by centrifugation, lysis is performed with Laemmli buffer and boiling. Lysed cells are centrifuged, and the soluble proteins from the supernatant are separated by SDS-PAGE or Phos-tag SDS-PAGE and immunoblotted with anti phospho-antibodies or anti-GST/His antibodies, respectively. The figure was partly generated using Servier Medical Art, provided by Servier, licensed under a Creative Commons Attribution 3.0 unported license.

**Figure 2 ijms-25-03813-f002:**
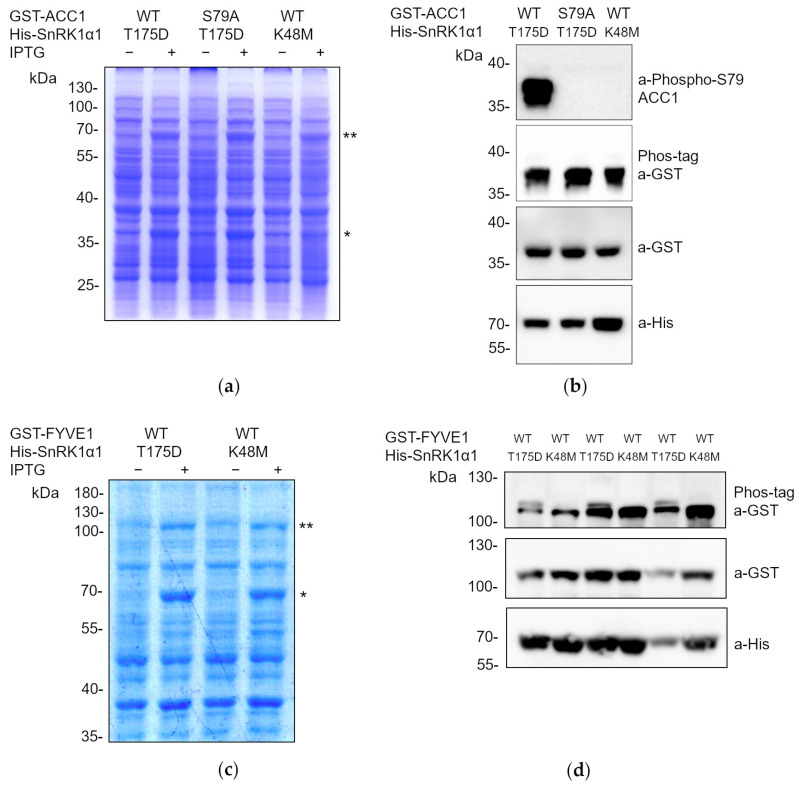
Constitutively active SnRK1α1 phosphorylates ACC1 and FYVE1 *in E. coli* cells. (**a**,**c**) Coomassie staining of total protein extracts (15 µL) separated in an 8% (**b**) or 10% (**a**) SDS-PAGE, obtained from *E. coli* expressing the indicated proteins upon IPTG induction. One or two asterisks in (**a**) indicate the position in the gel of the expression of GST-ACC1 and His-SnRK1α1, respectively. One or two asterisks in (**c**) indicate the position in the gel of the expression of His-SnRK1α1 and GST-FYVE1, respectively. (**b**,**d**) WB analyses, after standard or Phos-tag SDS-PAGE, of the indicated *E. coli* total protein extracts (5 µL) induced by IPTG with the described antibodies. Three independent IPTG inductions for each construct are shown in (**d**). His-SnRK1α1^T175D^, constitutively active; His-SnRK1α1^K48M^, constitutively inactive; GST-ACC1^S79A^, non-phosphorylatable S79.

**Figure 3 ijms-25-03813-f003:**
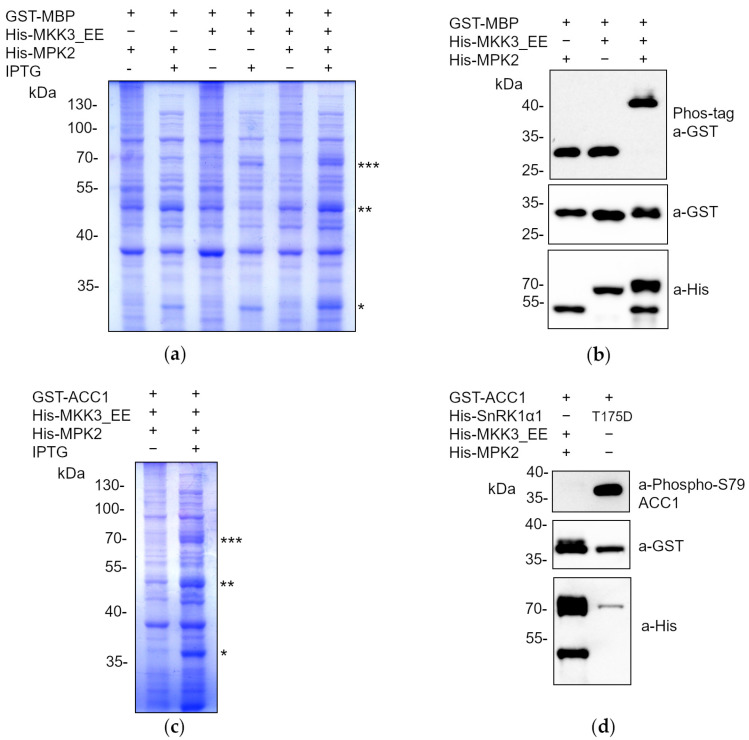
The MKK3-MPK2 regulation module phosphorylates MBP in *E. coli* cells. (**a**,**c**) Coomassie staining of total protein extracts (15 µL) separated in a 10% SDS-PAGE, obtained from *E. coli* expressing the indicated proteins upon IPTG induction. One, two, and three asterisks in (**a**) indicate the position in the gel of the expression of GST-MBP, His-MPK2, and His-MKK3, respectively. One, two, and three asterisks in (**c**) indicate the position in the gel of the expression of GST-ACC1, His-MPK2, and His-MKK3, respectively. (**b**,**d**) WB analyses with the described antibodies, after standard or Phos-tag SDS-PAGE, of the indicated *E. coli* total protein extracts (5 µL) induced by IPTG. His-MKK3^EE^, constitutively active; His-SnRK1α1^T175D^, constitutively active.

**Table 1 ijms-25-03813-t001:** List of entry vectors used in this work.

Name	Source
pENTR 3C-KIN10	Belda-Palazón et al. (2012) [54]
pDONR/ZEO-KIN10_T175D	This work
pDONR_ZEO-ACC_S55-V108	This work
pDONR_ZEO-ACC_S55-V108_S79A	This work
pDONR207-MKK3_S235E_T241E	Danquah et al. (2015) [33]
pDONR/ZEO-MPK2	This work
pDONR_ZEO-MBP_V93-Q102	This work

**Table 2 ijms-25-03813-t002:** List of primers used in this work.

Name	Sequence
MPK2_attB1	GGGGACAAGTTTGTACAAAAAAGCAGGCTTCATGGCGACTCCTGTTGAT
MPK2_attB2	GGGGACCACTTTGTACAAGAAAGCTGGGTATCAAAACTCAGAGACCTCATT
MBP_V93-Q102_attB1	GGGACAAGTTTGTACAAAAAAGCAGGCTTCGTGACCCCGCGCACCCCGCCGCCGAGCCAG
MBP_V93-Q102_attB2	GGGACCACTTTGTACAAGAAAGCTGGGTACTGGCTCGGCGGCGGGGTGCGCGGGGTCAC
ACC1_S55-V108_attB1	GGGGACAAGTTTGTACAAAAAAGCAGGCTTCATGTCAGATACACTTTCTGATTT
ACC1_S55-V108_attB2	GGGGACCACTTTGTACAAGAAAGCTGGGTAAACAAATTCTGCTGGCGAAGC
KIN10_Fw_attB1	GGGGACAAGTTTGTACAAAAAAGCAGGCTTCATGGATGGATCAGGCACAGG
KIN10_Rv_attB2	GGGGACCACTTTGTACAAGAAAGCTGGGTATCAGAGGACTCGGAGCTGAGCA
KIN10_K48M_Rv	CGACGATTGAGGATCATGATAGCAACCTTAT
KIN10_K48M_Fw	ATAAGGTTGCTATCATGATCCTCAATCGTCG
KIN10_T175D_Rv	GGACTTCCACAACTATCCTTCAAAAAATGAC
KIN10_T175D_Fw	GTCATTTTTTGAAGGATAGTTGTGGAAGTCC

**Table 3 ijms-25-03813-t003:** List of destination vectors generated in this work.

Name	Cloning Method
pQLinkH-KIN10_T175D	LR reaction: linearized pDONR/ZEO-KIN10_T175D + pQLinKH
pQLinkH-KIN10_K48M	LR reaction: linearized pDONR/ZEO-KIN10_K48M + pQLinKH
pQLinkG-ACC_S55-V108	LR reaction: linearized pDONR/ZEO-ACC_S55-V108D + pQLinKG
pQLinkG-ACC_S55-V108_S79A	LR reaction: linearized pDONR/ZEO-ACC_S55-V108D_S79A + pQLinKG
pQLinkG-MBP_V93-Q102	LR reaction: linearized pDONR/ZEO-MBP_V93-Q102 + pQLinKG
pQLinkH-MKK3_S235E_T241E	LR reaction: linearized pDONR/ZEO-MKK3_S235E_T241E + pQLinKH
pQLinkH-MPK2	LR reaction: linearized pDONR/ZEO-MPK2 + pQLinKH
pQLinkH-KIN10_T175D + G-ACC_S55-V108	LIC reaction: SwaI fragment pQLinkH-KIN10_T175D + PacI fragment G-ACC_S55-V108
pQLinkH-KIN10_T175D + G-ACC_S55-V108_S79A	LIC reaction: SwaI fragment pQLinkH-KIN10_T175D + PacI fragment G-ACC_S55-V108_S79A
pQLinkH-KIN10_K48M + G-ACC_S55-V108	LIC reaction: SwaI fragment pQLinkH-KIN10_K48M + PacI fragment G-ACC_S55-V108
pQLinkH-MPK2 + G-MBP_V93-Q103	LIC reaction: SwaI fragment pQLinkH-MPK2 + PacI fragment G-MBP_V93-Q103
pQLinkH-MKK3_S235E_T241E + G-MBP_V93-Q104	LIC reaction: SwaI fragment pQLinkH-MKK3_S235E_T241E + PacI fragment G-MBP_V93-Q104
pQLinkH-MPK2 + H-MKK3_S235E_T241E + G-MBP_V93-Q105	LIC reaction: SwaI fragment pQLinkH-MPK2 + PacI fragment H-MKK3_S235E_T241E + G-MBP_V93-Q105
pQLinkH-MKK3_S235E_T241E + G-ACC_S55-V108	LIC reaction: SwaI fragment pQLinkH-MKK3_S235E_T241E + PacI fragment G-ACC_S55-V108
pQLinkH-MPK2 + H-MKK3_S235E_T241E + G-ACC_S55-V108	LIC reaction: SwaI fragment pQLinkH-MPK2 + PacI fragment H-MKK3_S235E_T241E + G-ACC_S55-V108

**Table 4 ijms-25-03813-t004:** List of antibodies used in this work.

Name	Source	Animal	Dilution
Anti-GST	GeneCopoeia (Rockville, MD, USA) CGAB-GST-0050	Mouse	1:2000
Anti-His	Merck (Darmstadt, Alemania) 70796-3	Mouse	1:2000
Anti-Phospho-S79-ACC1	Cell Signaling (Danvers, MA, USA) #1673661S	Rabbit	1:1000
Anti-rabbit IgG-HRP	Jackson ImmunoResearch (West Grove, PA, USA) #111035144	Goat	1:20,000
Anti-mouse IgG-HRP	Jackson ImmunoResearch (West Grove, PA, USA) #115035146	Goat	1:20,000

## Data Availability

All data supporting the findings of this study are available in the main text. Additional data related to this study are available from the corresponding author upon request. All biological materials used in this study are available from the corresponding authors on reasonable request.

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
