# Peer review of "An Escherichia coli-Based Phosphorylation System for Efficient Screening of Kinase Substrates"

_ijms, 2024, doi:10.3390/ijms25073813_

Round 1

Reviewer 1 Report

Comments and Suggestions for Authors

This manuscript presents a methodology to monitor phosphorylation events using E. coli as a heterologous system. Indeed, this work presents a relatively simple and efficient phosphorylation detection system that can be very useful for those scientists who does not have an opportunity to use radioactive approaches or others for phosphorylation event detection. Thus, the topic of this manuscript is interesting and relevant for IJMS. I recommend this Ms for publication in IJMS after a revision.

 Specific comments:

1) The legend to the Figure 1 should be considerably extended with the brief enumeration and description of the main methodological steps.

2) It is important to include in the Introduction a brief description what are the current available experimental method of plant kinase substrate detection/screening in the world practice. Briefly, mention what are advantages and disadvantages of these approaches.

3) It is also necessary to mention in the Introduction the aim of this study.

4) In the Introduction, it is also important to explain why your approach is unique and novel compared to other phosphorylation detection methods.

5) I recommend that an additional Figure should be added that schematically shows what methods exist for detecting phosphorylation in plants and how they are relate to each other.

Author Response

Comments to REVIEWER 1

1) The legend to the Figure 1 should be considerably extended with the brief enumeration and description of the main methodological steps.

According to the suggestion of the reviewer, we have detailed the methodology in the legend to the Figure 1.

2) It is important to include in the Introduction a brief description what are the current available experimental method of plant kinase substrate detection/screening in the world practice. Briefly, mention what are advantages and disadvantages of these approaches.

We agree with both reviewers the lack of information of our submitted manuscript, with regard to the contribution of the Phoscreen technique to current available methodology to study protein phosphorylations. Therefore, we have modified both the Introduction and Discussion sections in the manuscript to properly place this technology in the context of other related methods.

3) It is also necessary to mention in the Introduction the aim of this study.

Following the reviewer´s suggestion, we have emphasized in the Abstract, and throughout the text the aim of the study.

4) In the Introduction, it is also important to explain why your approach is unique and novel compared to other phosphorylation detection methods.

In line with the reviewer's recommendation, we have improved the description of the role that this technology can play as a complement to currently used methods, in particular to in vitro phosphorylation assays.

5) I recommend that an additional Figure should be added that schematically shows what methods exist for detecting phosphorylation in plants and how they are related to each other.

We appreciate the reviewer's suggestion, and in response, we have incorporated a comprehensive discussion of the pros and cons of our method compared to particularly in vitro phosphorylation assays. After careful consideration, we have chosen not to make substantial changes to the manuscript. Consequently, we have opted against creating an additional figure comparing existing methods for detecting phosphorylation in plants. Currently, large-scale phosphorylation studies are conducted through phospho-proteomic analyses, and in no way does the method developed in this work aim to replace such extensive studies. Following these unbiased analyses (or, alternatively, after performing in silico analyses of potential phosphorylation sites), the confirmation of potential phosphorylation candidates is primarily carried out through in vitro phosphorylation assays. For this validation, the method developed in this work represents an efficient alternative for tracing these potential candidates. As the aim of this work is not to review the existing methods for detecting phosphorylation in plants and this could dilute the main message of the work, we have alternatively chosen to update the bibliography by incorporating new and relevant citations to enhance the scholarly references (cites 11-14).

Reviewer 2 Report

Comments and Suggestions for Authors

Cayuela et al. describe a nice system for monitoring particular phosphorylation events. They co-express a kinase and its putative substrate in E.coli cells, prepare lysates and test substrate’s phosphorylation by anti-phospho-specific antibodies, if available, or by anti-substrate antibody in gels containing Phos-tag.

A couple of roofs-of-concept for the validity of the methods are provided, and include all necessary positive and negative controls. One of the system include a cascade of MAP kinases. Overall the experiments are performed professionally, the results are convincing and description of the method is fluent.

While the method (rather the trick) described could be useful for specific kinase-substrate studies, it is not of unusual value and certainly cannot replace in vitro kinase assays. Indeed the method saved the need of protein purification, but lack much of the qualities of in vitro assays, primarily radioactive assays. Examples: the method is not really quantitative, does not allow determining catalytic and kinetic parameters and is no open to testing inhibitors, allosteric effectors etc. Also, unlike their claim, the system is not permissive for a real high throughput screening. Finally, although saving the need of protein purification it requires cloning and using several controls, so it does not really reduces effort. I strongly recommend the authors to soften significantly their statements and language regarding the new method. It is a very nice trick, nicely executed, but far from being revolutionary.

The Introduction and Discussion are too long and carried away to non-relevant matters. The method is universal for any kinase, not necessarily plant kinases, and certainly not specific for the kinases used as examples. The description of these kinases and their role in plant metabolism is totally dispensable and in fat dilute the message. The paper should be of about half the size it is now.

Author Response

Comments to REVIEWER 2

While the method (rather the trick) described could be useful for specific kinase-substrate studies, it is not of unusual value and certainly cannot replace in vitro kinase assays. Indeed the method saved the need of protein purification, but lack much of the qualities of in vitro assays, primarily radioactive assays. Examples: the method is not really quantitative, does not allow determining catalytic and kinetic parameters and is no open to testing inhibitors, allosteric effectors etc. Also, unlike their claim, the system is not permissive for a real high throughput screening. Finally, although saving the need of protein purification it requires cloning and using several controls, so it does not really reduces effort. I strongly recommend the authors to soften significantly their statements and language regarding the new method. It is a very nice trick, nicely executed, but far from being revolutionary.

We agree with the reviewer that our submitted manuscript lacked a proper integration of the methodology Phoscreen in the context of related available methodology. It is true that the Phoscreen technique provides some advantages but also drawbacks compared for instance to in vitro radioactive assays mostly with regard to studies of enzyme kinetics. Therefore, we have modified the sections Abstract, Introduction and Discussion to properly discuss the role of the Phoscreen as a complement to the current catalogue of related methods.

The Introduction and Discussion are too long and carried away to non-relevant matters. The method is universal for any kinase, not necessarily plant kinases, and certainly not specific for the kinases used as examples. The description of these kinases and their role in plant metabolism is totally dispensable and in fat dilute the message. The paper should be of about half the size it is now.

We agree with the reviewer that the manuscript submitted included a too extensive introduction of the kinases selected to validate the Phoscreen technique taking into account the universality of the method. Therefore, we have shortened large paragraphs dedicated to this, mostly in the Introduction section, to have a more balanced introductory section and to reduce the extension of the manuscript according to the reviewer suggestion. Nevertheless, we have chosen to retain a portion of the description regarding the function of these kinases. Selection of these kinases for the development of this technology is of great interest to the plant scientific community, given their significant role in plant stress response.